# Microbiome Responses to Oral Fecal Microbiota Transplantation in a Cohort of Domestic Dogs

**DOI:** 10.3390/vetsci11010042

**Published:** 2024-01-19

**Authors:** Connie A. Rojas, Zhandra Entrolezo, Jessica K. Jarett, Guillaume Jospin, Alex Martin, Holly H. Ganz

**Affiliations:** AnimalBiome, Oakland, CA 94609, USA; connie@animalbiome.com (C.A.R.); zhandra@animalbiome.com (Z.E.); jess@animalbiome.com (J.K.J.); guillaume@animalbiome.com (G.J.); alex@animalbiome.com (A.M.)

**Keywords:** fecal microbiota transplant, oral capsule FMTs, fecal microbiome, domestic dogs, 16S rRNA gene sequencing, diarrhea, antibiotics, kibble, raw food

## Abstract

**Simple Summary:**

We report on the changes observed in the microbiomes of fifty-four dogs that received fecal transplants (FMTs) in the form of oral capsules for their chronic vomiting, diarrhea, and/or constipation. We found that the relative abundances of short-chain fatty acid producing bacteria increased after FMT. The microbiome compositions of dogs before and after FMT were associated with the diet and antibiotic history. Furthermore, we found that certain groups of donor bacteria were more commonly shared with the FMT recipient. Lastly, our data suggested that a high degree of overlap between the microbiome of the donor and that of the recipient was negatively associated with bacterial engraftment, and could be an important factor to consider when evaluating the impact of FMTs on the host and its microbiome.

**Abstract:**

Fecal microbiota transplants (FMTs) have been successful at treating digestive and skin conditions in dogs. The degree to which the microbiome is impacted by FMT in a cohort of dogs has not been thoroughly investigated. Using 16S rRNA gene sequencing, we document the changes in the microbiome of fifty-four dogs that took capsules of lyophilized fecal material for their chronic diarrhea, vomiting, or constipation. We found that the relative abundances of five bacterial genera (*Butyricicoccus*, *Faecalibacterium*, *Fusobacterium*, *Megamonas*, and *Sutterella*) were higher after FMT than before FMT. Fecal microbiome alpha- and beta-diversity were correlated with kibble and raw food consumption, and prior antibiotic use. On average, 18% of the stool donor’s bacterial amplicon sequence variants (ASVs) engrafted in the FMT recipient, with certain bacterial taxa like *Bacteroides* spp., *Fusobacterium* spp., and *Lachnoclostridium* spp. engrafting more frequently than others. Lastly, analyses indicated that the degree of overlap between the donor bacteria and the community of microbes already established in the FMT recipient likely impacts engraftment. Collectively, our work provides further insight into the microbiome and engraftment dynamics of dogs before and after taking oral FMTs.

## 1. Introduction

Chronic enteropathy (CE) is a prevalent intestinal inflammatory disorder in dogs, characterized by persistent and/or recurrent vomiting, diarrhea, decreased appetite, abdominal pain, and weight loss lasting longer than three weeks [1]. The condition profoundly affects the dog’s health and quality of life. Fortunately, several forms of treatment are available for CE, which may involve dietary changes [2], prebiotics and probiotics [3,4,5], antibiotics [6,7], or steroids [8,9]. The course of treatment depends on whether the enteropathy is classified as antibiotic-responsive (ARE), immunosuppressant-responsive (IRE), food-responsive (FRE), or non-responsive (NRE) [1,10,11]. Yet another mode of treatment that shows promise in treating CE are fecal microbiota transplants (FMTs) [12]. 

FMTs involve the transfer of fresh or freeze-dried fecal material from a healthy donor into the gastrointestinal tract of a recipient in the form of endoscopies, rectal enemas, or oral capsules [12,13]. In dogs, fecal microbiota transplants have been effective at resolving clinical signs in individuals with acute (hemorrhagic) diarrhea [14,15], parvovirus-associated diarrhea [16], *Clostridium perfringens*-associated diarrhea [17], antibiotic-responsive or non-responsive enteropathy [17,18,19,20,21,22,23], and canine atopic dermatitis [24,25]. Fecal transplants aim to restore the gut microbiome by increasing microbiome diversity, repopulating numbers of beneficial bacteria and their metabolites, or reducing the abundances of potential pathogens. In one study, for example, dogs with acute hemorrhagic diarrhea receiving a single colonoscopic FMT experienced increases in the relative abundances of commensal or beneficial bacteria including *Eubacterium biforme*, *Porphyromonas*, *Megamonas*, *Megasphaera*, *Prevotella* copri, and *Peptococcus,* compared to saline FMT controls [14]. In another study of a seven-year-old dog with NRE, twenty-two bacterial genera including *Fusobacterium*, *Sutterella*, *Megamonas*, and *Peptoclostridium* were introduced as a result of a single endoscopic FMT and remained in the gut seven months after FMT treatment [17].

In other cases, large changes in the microbiome may not be observed or be required for FMT effectiveness. A recent study administered FMTs in the form of oral capsules to twenty-seven dogs with CE [23] and found that 17 days after treatment, the fecal microbiomes of recipients remained distinct from the fecal microbiomes of healthy dogs and were less diverse. However, FMT recipients did generally exhibit declines in their dysbiosis indices and improvement in their clinical signs. Several factors could be modulating how FMT recipients and their microbiomes respond to FMT, such as prior diagnoses, prior antibiotic exposure, or the dog’s diet and lifestyle. For example, a recently published study found that microbiome responses to oral FMTs in cats with chronic digestive issues were individual-specific and correlated with host clinical signs and diet [26]. Nonetheless, this type of information is unknown for dogs.

The composition of a donor’s fecal microbiome is also an important factor to consider when evaluating responses to FMT. Prior work demonstrates that the microbiome of the FMT recipient may become more similar to that of the stool donor and more diverse, as was observed for a nine-year-old dog suffering from CE [18]. The dog showcased increases in the relative abundances of *Fusobacteriaceae*, *Bacteroidaceae*, *Prevotellaceae*, *Ruminococcaceae*, *Veillonellaceae* and *Erysipelotrichaceae* so they mirrored the donor’s relative abundances. Certain groups of bacterial taxa may also be more likely to be transferred from stool donors to their recipients. In cats receiving oral FMTs for chronic digestive issues [26], the most commonly engrafted ASVs belonged to bacteria in the genera *Peptoclostridium*, *Bacteroides*, *Prevotella*, and *Collinsella*. In humans, strain engraftment for patients with recurrent *C. difficile* (rCDI) infection was impacted by the abundance and phylogenetic breadth of both the donor’s bacteria and the FMT recipient’s bacteria [27]. To date, no studies have investigated which bacteria engraft and what degree of engraftment is observed in a cohort of dogs receiving FMTs.

Here, we address some of these gaps in knowledge and examine the fecal microbiome responses to oral capsule FMTs in a cohort of fifty-four dogs with chronic vomiting, diarrhea, or constipation lasing more than 2 weeks (Table 1). The dogs had clinical signs consistent with CE but did not all meet the diagnostic criteria for a formal diagnosis. We document the changes that were observed at the microbiome level after FMT and correlate these microbiome responses to five host factors of interest: host clinical signs, raw food consumption, dry food consumption, prior antibiotic use, and body condition score. We also compare the fecal microbiomes of FMT recipients to their fecal donors (*n* = 7) and examine which microbes ‘engrafted’ in the recipient and the proportion of the microbiome they constituted. Collectively, our study provides a detailed analysis of the microbiome changes exhibited by dogs after receiving a 25-day course of oral capsule FMTs. 

## 2. Methods

### 2.1. FMT Participants

Fifty-four dogs with diarrhea, vomiting, or constipation lasting more than two weeks were recruited for this study using social media. After signing a consent form, pet owners received a bottle containing 50 FMT capsules, a health survey, and gloves and tubes (containing 70% ethanol and silica beads) to collect fecal samples. A fecal sample was collected prior to the start of the study and two weeks after the end of the FMT capsule course. Owners were instructed to give two capsules daily to their dog orally with food for a duration of 25 days. All dogs must have taken all 50 FMT capsules to be included in this study. Fecal samples were shipped to AnimalBiome (Oakland, CA, USA) and stored at 4 °C until genomic DNA extraction. 

Owners provided information on their dog’s age, sex, body condition, breed, spay or neuter status, diet, and any clinical conditions or diagnoses given by veterinarians (Appendix A). Although the dogs did not take antibiotics during FMT treatment, 65% of the dogs had had a prior antibiotic exposure at some point during the 12 months preceding the study. For most dogs, their diet did not change during the study period.

### 2.2. Preparation of FMT Capsules

To make the FMT capsules, fecal samples were collected from twelve healthy dog donors (n = 12). The fecal samples were subsequently screened for a range of parasites and pathogens including *Cryptosporidium* spp., *Clostridioides difficile* toxins A and B, *Giardia* spp., *Salmonella* spp., and *Tritrichomonas foetus* using both qPCR and culturing. This work was performed at the University of California, Davis Real-time PCR and Diagnostics Core Facility (Davis, CA, USA). To qualify as donors, dogs had to fulfill the following: no antibiotic treatment in the past year; not be taking medications; not have any known health conditions; no current infections or recent surgeries; and no behavioral issues. Donors were required to test negative for protozoan oocysts and helminth parasites (IDEXX, Westbrook, ME, USA). The donors in our study had a median age of 4.04 years (range: 1–9 years old), were 58% male, mostly spayed or neutered (75%), and with body condition scores between 4 and 6 (inclusive) (Appendix A). They represented seven distinct breeds, among them PitBull, Border Collie, and Poodle mixes.

### 2.3. DNA Extractions and Illumina 16S rRNA Gene Sequencing

Total DNA was extracted from fecal samples of 54 FMT recipients (108 fecal samples) and 12 donors (22 fecal samples) using the QIAGEN DNeasy PowerSoil Kit (Germantown, MD, USA). Amplification of the 16S rRNA gene (V4) was achieved using a dual-indexing one-step PCR with the 505F/816R primers (Integrated DNA Technologies, Coralville, IA, USA) and multiple barcodes as outlined in Pichler et al. 2018 [28]. The PCR mix contained 0.3–30 ng of template DNA, 0.2 mM of each dNTP, 0.1 µL Phusion DNA Polymerase (ThermoFisher, Waltham, MA, USA), 1X HF PCR Buffer, and 10 µM of each primer. PCR products were purified using the SequalPrep Normalization Kit (Thermo Fisher) and pooled into the final libraries; each contained 95 samples (not all from this study) and at least one positive control and one “no template” control. To quantify the final libraries, we used the QUBIT dsDNA high-sensitivity (HS) assay (Thermo Fisher, Waltham, MA, USA). These were diluted to 1.8 pM and denatured in preparation for paired-end sequencing (150 bp) on the Illumina MiniSeq. 

### 2.4. Amplicon Sequence Processing in DADA2

Sequences generated from the Illumina platform were imported into R (v4.3.0) and truncated, quality-filtered and dereplicated using the Divisive Amplicon Denoising Algorithm (DADA2 v1.14.1) [29,30]. Prior to calculating error rates, we trimmed both forward and reverse reads to 145 bp. Sequences were then denoised to infer amplicon sequence variants (ASVs)—the most finely resolved measure of taxonomy we have available. Chimeras were identified and subsequently removed. Post processing, samples from FMT recipients had a median of 60,145 sequences and those from donors had a median of 66,870 sequences. Taxonomic annotation of ASV sequences was accomplished with the Silva reference database (v138) [31,32], setting a minimum bootstrap confidence threshold of 80%. ASVs given a taxonomic label of Mitochondira, Chloroplasts, or Eukarya were removed, as were ASVs that were unclassified at the domain level. The table of ASV counts, list of ASV taxonomic assignments, and sample metadata were saved for statistical analyses, and are available as Appendix A.

### 2.5. Statistical Analysis of Microbiome Data

All statistical analysis and figures pertaining to this study were performed in the R statistical software program (v.4.3.0).

Microbiome composition. To visualize microbiome composition, we plotted the relative abundances of bacterial genera before and after FMT in the form of stacked bar plots. Bacterial genera with average relative abundances >1.3% were displayed and all others were clumped into an “Other” category. Dog names were anonymized.

The LinDA R package (v0.1.0) [33] was used to test whether the relative abundances of bacterial genera differed between pre-FMT and post-FMT samples. The LinDA model included sample type (before FMT vs. after FMT) as a main predictor and dog name as a random effect to account for repeated measures from the same individual. The prevalence cutoff was set to 20%, winsorization cutoff (quantile) to 0.97, and *p*-value adjustment to “FDR”. Results were visualized with boxplots using CLR transformed counts to match the log-2 transformation applied in the analysis.

Alpha-diversity. For alpha-diversity analyses, samples were first subsampled to 26,000 sequences (GUniFrac package (v1.7) [34]) to control for sequencing depth. Six samples (two pre-FMT and four post-FMT) had fewer sequences than this cutoff and were excluded from all alpha-diversity analyses. Three metrics of microbiome alpha-diversity were computed with the phyloseq package v1.44.0 [35]. The metrics were Chao 1 Richness, Shannon Diversity, and Gini-Simpson Evenness (1-Simpson’s index). A linear mixed-effects model with a Gaussian distribution tested whether logged Chao 1 richness values were different between pre-FMT and post-FMT samples, setting dog identity as a random effect. Generalized linear mixed-effects models with a Gamma distribution tested whether Shannon Diversity or Gini-Simpson evenness differed between pre-FMT and post-FMT samples, specifying the same random effect. Model outputs were inspected for each analysis (e.g., qqplots) to ensure normality of residuals.

Another set of generalized linear models correlated the three metrics of microbiome alpha-diversity with the five host predictors of interest: clinical signs (Diarrhea only, Vomiting only, Diarrhea with Vomiting, any Constipation), raw food consumption (Y/N), kibble consumption (Y/N), prior antibiotic use (Y/N), and body condition score (numeric). These linear mixed effects models were conducted using lme4 (v1.1-34) [36]. Post hoc tests were conducted using the emmeans (v1.8.7) [37] and multcomp (1.4-23) [38] packages and P-values were adjusted using Tukey’s method. Estimated marginal means with 95% confidence intervals were extracted. 

Beta-diversity. For beta-diversity analyses, we computed three dissimilarity distances with the vegan package: Jaccard distance based on the presence/absence of ASVs, Bray–Curtis distance based on ASV proportions, and Aitchison distances based on CLR-transformed ASV counts. These served as input for Permutational Multivariate Analyses of Variance (PERMANOVAs; 1000 permutations). One PERMANOVA model tested whether fecal microbiome beta-diversity differed between pre-FMT and post-FMT samples. Another model tested whether fecal microbiome beta-diversity was significantly associated with the five host predictors of interest. Pairwise comparisons (e.g., post hoc tests) were conducted with the pairwise Adonis package (v0.4.1) [39]. 

Donor bacteria engraftment. We calculated bacterial engraftment rates by dividing the number of ASVs in common between postFMT samples and their stool donors (excluding ASVs shared between pre-FMT samples and donors) by the total number of ASVs in the donor sample (excluding ASVs in common between pre-FMT samples and donors). Higher rates would indicate that most of the donor’s ASVs were shared with the FMT recipient. To calculate this, we first filtered the dataset to remove singleton and doubleton ASVs (ASVs with a summed count of 1 or 2 reads in the entire dataset); these could otherwise inflate or misconstrue ASV engraftment rates.

Generalized linear models were used to identify the host or donor factors that were associated with ASV engraftment rates. One model regressed logged ASV engraftment rates with clinical signs, raw food consumption, kibble consumption, prior antibiotic use, and body condition score. Another model correlated these same engraftment rates with donor identity. A third model regressed ASV engraftment rates with the microbiome alpha-diversity of the donor or the FMT recipient. All generalized linear models were constructed using the stats package (v4.3.0) [29,30]. 

We examined ASV engraftment from the perspective of the FMT recipient as well. Essentially, we quantified the recipient’s microbiome that was derived from ASVs of the donor, those of the recipient preFMT, those that were shared between donor and recipient from the beginning, or those that were environmental/stochastic (e.g., did not come from recipient or donor). Higher engraftment would mean that a large proportion of the recipient’s microbiome contained donor-derived ASVs.

Community ecology dictates that the recipient’s starting microbiome can influence the type of donor microbes that can establish themselves after FMT [40]. Thus, we ran Spearman correlations to compare the summed abundances of donor-derived ASVs (e.g., proportion of the microbiome they constituted) with the degree of overlap/similarity between the donor’s microbiome and recipient’s pre-FMT microbiome. Microbiome similarity between donor–recipient pairs were computed using Unifrac distances based on ASV counts and a phylogenetic tree of ASVs made with DECIPHER (v2.14.0) [41] and phangorn (v2.5.5) [42]. 

## 3. Results

### 3.1. Description of Dog Cohort

Fifty-four dogs received a full course of FMT oral capsules for their chronic diarrhea, vomiting, or constipation. All dogs had clinical signs consistent with CE but did not have a formal diagnosis or meet all of the criteria for a formal diagnosis. Participants had a median age of 5.2 years, and a median body condition score of 5 (range: 2–8). About a quarter were under 22 lbs, another quarter were between 40–60 lbs, and the remaining weighed over 60 lbs (Table 1). There were slightly more males (54%) than females (46%). Over 20 unique breeds were represented, and among the most common were Poodles, Golden Retrievers, and Terriers (Table 1). Seventy-eight percent were spayed or neutered and 65% had a prior antibiotic exposure (within the year preceding sample collection). Regarding clinical signs, 48% were suffering from diarrhea, 30% had episodes of vomiting with diarrhea, 13% experienced only vomiting, and 9% exhibited signs of constipation (e.g., constipation with vomiting, constipation with vomiting and diarrhea).

### 3.2. The Composition of Canine Fecal Microbiomes before and after FMT

Before FMT, the fecal microbiomes of recipients were characterized by high abundances of *Escherichia* (11.07% mean relative abundance), *Bacteroides* (9.16%), *Fusobacterium* (8.23%), *Streptococcus* (6.74%), *Prevotella* 9 (5.44%), and *Blautia* (5.55%). But the relative abundances of these taxa did shift after FMT. Some dogs experienced decreases in their *Streptococcus* or *Escherichia* abundances, and others demonstrated increases in their *Blautia* or *Fusobacterium* abundances (Figure 1). However, for other dogs, dramatic shifts in the microbiome were not observed (Figure 1). This suggests that microbiome responses to FMT are not uniform and vary across individuals.

Similar patterns were observed when analyzing microbiome beta-diversity. Ordination plots showed that a handful of recipients did undergo noticeable shifts in their microbiome after FMT (Figure 2A) while others did not exhibit much change (e.g., there was little distance between their pre-FMT and post-FMT samples). Statistical analyses supported these observations and indicated that microbiome responses were individual-specific, with host identity accounting for 69% of the variation (PERMANOVA Jaccard R^2^ = 0.64, *p* = 0.009; Bray–Curtis R^2^ = 0.69, *p* = 0.0009; Aitchison R^2^ = 0.67, *p* = 0.0009). Regardless of the degree to which each recipient’s microbiome changed after FMT, differential abundance analyses revealed that the relative abundances of *Butyricicoccus*, *Faecalibacterium*, *Fusobacterium*, *Megamonas*, and *Sutterella* were higher in post-FMT samples than in pre-FMT samples (LinDA *p* < 0.05, Figure 2B, Appendix A). 

Interestingly, fecal microbiome alpha-diversity before FMT was not different from diversity after FMT (Chao 1 Richness LMM β = −0.08 ± 0.05, *p* = 0.12; Shannon diversity GLMM β = 0.01 ± 0.01, *p* = 0.33; Gini-Simpson GLMM β = −0.21 ± 0.4, *p* = 0.59).

### 3.3. Host Factors Associated with Microbiome Alpha- and Beta-Diversity

Prior to FMT, canine fecal microbiomes were significantly associated with diet components, recent antibiotic use, and body condition scores (GLM *p* < 0.05; see Appendix A for full statistical output). Dogs that consumed dry food had more rich microbiomes (mean Chao 1 Richness 104.9) than dogs that did not (mean Chao 1 Richness 88.8); similarly, dogs that consumed raw food had more rich microbiomes (mean Chao 1 Richness 103.6) than dogs that did not (mean Chao 1 Richness 87.04) (Figure 3A,B). Furthermore, dogs with a prior antibiotic exposure had slightly more diverse microbiomes (mean Shannon 2.7) than dogs without a recent exposure (mean Shannon 2.4) (Figure 3C). Lastly, dogs with larger body condition scores had greater diversity than dogs with lower body condition scores (Gini-Simpson GLM β = 0.016 ± 0.008, *p* = 0.04) (Figure 3D). Clinical signs were not significantly predictive of fecal microbiome diversity (*p* > 0.05, Appendix A). After FMT, none of the host factors predicted fecal microbiome alpha-diversity (Table 2 and Appendix A).

For beta-diversity, the fecal microbiomes of recipients before FMT were structured by diet and prior antibiotic use (PERMANOVA *p* < 0.05; Figure 4; Appendix A); each factor explained 3–5% of the variation in the microbiome. Specifically, the fecal microbiomes of dogs that ate kibble were different from the microbiomes of dogs that did not eat any kibble, and a similar pattern was observed for raw food (Figure 4A–E; Appendix A). Differential abundance testing revealed that dogs fed raw food were enriched in *Bacteroides*, *Collinsella*, *Slackia*, and *Fusobacterium* (Figure 4B; Appendix A) compared to dogs that did not consume raw food. No bacteria were identified as being differentially abundant in dogs fed kibble compared to dogs not fed kibble. Dogs without a recent antibiotic exposure were enriched in *Allobaculum*, *Fusobacterium*, *Meganomas*, *Peptoclostridium*, and *Peptococcus* compared to dogs with an antibiotic exposure which instead had more *Clostridioides* and *Streptococcus* (Figure 4C,D; Table 2 and Appendix A).

After FMT, fecal microbiome beta-diversity was significantly associated with kibble consumption and prior antibiotic use (Appendix A), and each factor accounted for ~3% of variation in the microbiome. Differential abundance testing was not able to identify any bacterial genera as accounting for differences in the microbiome between our different groups (LinDA results not shown because all adjusted *p* values > 0.05). 

### 3.4. Bacterial Engraftment after FMT

We found that 2.63% to 62.12% of the stool donor’s ASVs engrafted in the FMT recipient (average: 18.29%, median: 15.34%) (Figure 5A; Appendix A). That is, of the bacterial ASVs present in the microbiomes of stool donors with the capacity to engraft (33–170 ASVs), about 18% on average (1–61 ASVs) successfully engrafted in FMT recipients (Appendix A). These rates were not significantly associated with clinical signs, dry kibble consumption, raw food consumption, prior antibiotic use, or body condition score (GLM LRT Clinical signs χ^2^ = 0.13, *p* = 0.98; Raw food χ^2^ = 0.91, *p* = 0.33; Dry food χ^2^ = 0.27, *p* = 0.59; Antibiotics χ^2^ = 0.07, *p* = 0.78; BCS χ^2^ = 1.34, *p* = 0.24). 

Additionally, ASV engraftment rates were not significantly predicted by donor identity (GLM LRT χ^2^ = 2.2, *p* = 0.82), but were modestly negatively correlated with the richness of the donors’ microbiome (GLM LRT Chao 1 χ^2^ = 3.3, *p* = 0.05; Shannon χ^2^ = 0.49, *p* = 0.48; Gini-Simpson χ^2^ = 0.63, *p* = 0.42) (Figure 5B). ASV engraftment rates were not correlated with the diversity of the recipient’s starting microbiome (GLM LRT Chao 1 χ^2^ = 0.19, *p* = 0.65; Shannon χ^2^ = 0.91, *p* = 0.34; Gini-Simpson χ^2^ = 0.29, *p* = 0.58). 

The most commonly engrafted ASVs were classified as uncultured *Lachnospiraceaea* (17.99% of all engrafted ASVs), *Lachnoclostridium* (6.9%), unclassified *Ruminococcaceae* (3.8%), *Blautia* (3.69%), *Ruminococcus torques* (3.69%), *Fusobacterium* (2.89%), and *Bacteroides* (2.76%) (Figure 5C). Bacterial taxa such as *Prevotella* 9, *Megamonas*, and *Alloprevotella* did not engraft as well (Figure 5C).

We also examined ASV engraftment from the perspective of the recipient. We compared the portion of their microbiomes that were composed of donor-derived ASVs to the portion composed of ASVs derived from the recipient or from the environment. We found that ASVs shared between donors and recipients pre-FMT made up the largest portion of the postFMT microbiome (~46% on average), followed by ASVs derived from the donor (20%) or the recipient (20%). Environmentally acquired (e.g., stochastic) ASVs made up the smallest fraction of the microbiome (~13% on average) (Figure 6B, Appendix A). Before FMT, ASVs shared between donors and recipients made up 62% of the microbiome, while ASVs unique to the recipient made up 38% (Figure 6A, Appendix A). Thus, it appeared that as new ASVs engrafted in the recipient, there was a larger reduction in recipient-derived ASVs (Wilcoxon rank-sum test pre vs. post W = 2011, *p* < 0.001) than ASVs originally shared between donors and recipients (Wilcoxon rank-sum test pre vs. post W = 1864, *p* < 0.003).

Interestingly, recipients whose starting microbiomes were very similar to their donors (weighted Unifrac similarity) tended to have less of their microbiome made up of donor ASVs than recipients with more dissimilar microbiomes to their donors (Spearman correlation *r* = −0.5, *p* = 0.0001) (Figure 6C).

## 4. Discussion

The primary focus of this study was to examine the microbiome responses of a cohort of dogs that received a 25-day course of oral capsule FMTs for their digestive issues. We found that the relative abundances of five bacterial genera—*Butyricicoccus*, *Faecalibacterium*, *Fusobacterium*, *Megamonas*, and *Sutterella*—increased after FMT. Microbiome alpha- and beta-diversity were best predicted by host diet and recent antibiotic use, and to a lesser extent, by body condition score. On average, 18% of the stool donor’s bacterial ASVs were transferred to the FMT recipient, and these rates were significantly associated with diversity of the donor’s microbiome. The most commonly engrafted bacterial groups included *Lachnospiraceaea*, *Lachnoclostridium*, *Blautia*, *Ruminococcus*, *Fusobacterium*, and *Bacteroides*. Lastly, we found that a high degree of similarity between the microbiome of the donor and its FMT recipient meant that a lesser portion of the recipient’s microbiome post-FMT was composed of donor-derived bacteria.

### 4.1. Changes in Canine Fecal Microbiomes after FMT

These results show that the microbiome composition of dogs that underwent a 25-day course of oral FMTs for chronic digestive issues exhibited increases in the relative abundances of five bacterial genera: *Butyricicoccus*, *Faecalibacterium*, *Fusobacterium*, *Megamonas*, and *Sutterella*. Similarly, dogs with acute hemorrhagic diarrhea that took FMTs via colonoscopy experienced increases in the relative abundances of thirty bacterial taxa, and among them were *Butyricicoccus pullicaecorum*, *Faecalibacterium prausnitzii*, and *Megamonas* [14]. Nine dogs with inflammatory bowel disease (IBD) that took FMTs in the form of rectal enemas also experienced increases in the relative abundances of *Fusobacterium* [21]. Three dogs with diarrhea showed increases in *F. prausnitzii* and *Blautia* spp. after administration of oral FMT capsules [43]. Interestingly, no other studies have reported an increase in *Sutterella* as a result of FMT in dogs, though this bacterial genus is less abundant in the microbiomes of dogs with IBD [44] or SRE [45] compared to healthy controls. According to several studies, *Faecalibacterium* and *Butyricicoccus* also increase in canine fecal microbiomes after the administration of probiotics [46,47].

Species from the genera *Butyricicoccus*, *Faecalibacterium*, *Fusobacterium*, and *Megamonas* are producers of short-chain fatty acids (SCFAs) like butyrate, acetate, or propionate [48,49,50], or of SCFA precursors like lactate [51]. This is significant given that SFCAs are important modulators of gut homeostasis, intestinal motility, and the immune system, and may have antidiarrheic effects. These gut commensals may also be involved in glycan biosynthesis and metabolism, the metabolism of cofactors and vitamins, and amino acid metabolism [47]. 

Our study did not find evidence for increased gut microbiome alpha-diversity diversity after FMT, contrasting findings from previous studies conducted in dogs that took FMTs for their atopic dermatitis [24], acute or hemorrhagic diarrhea [14,52], or chronic enteropathy [17,18]. However, the sample sizes for these studies were much smaller (1–11 dogs), used no statistics or different statistical tests, and employed other routes of FMT administration (e.g., enemas and endoscopies). The heterogeneity of the study populations could also be different. The dogs in the current study were of varying ages (1–15 years), breeds (20+ unique breeds), body condition scores (BCS 2–8), diets, and geographic locations. They had a plethora of underlying health conditions, and showcased different clinical signs. Other studies could have had a more narrowly defined group of dogs. The heterogeneity in our sample pool could have also contributed to the individualized microbiome responses showcased by FMT participants. No two FMT participants had identical preFMT or postFMT microbiomes. Similarly, a recently published study in cats also reported individual-specific microbiome responses to oral FMTs [26].

### 4.2. Microbiome Associations with Host Factors

The fecal microbiomes of FMT recipients before and after FMT were significantly associated with diet (kibble and/or raw food consumption), prior antibiotic use, and body condition score. 

It is widely known that host diet is a strong determinant of gut microbiome composition. The dietary macronutrients and substrates ingested by the host directly select for bacteria with specific metabolic capacities. The amount of fat, fiber, or protein content, digestibility, and palatability of a dog’s diet will have cascading effects on the microbiome, microbial metabolites, and microbial interactions like cross-feeding relationships. A fiber-rich (inulin) diet for example, enriches the microbiome in *Megamonas* spp. and *Lactobacillus* spp. [53]. A high-protein (red meat) diet favors the growth of *Fusobacterium*, *Lactobacillus*, and *Clostridium* [54]. A high-fat (33% fat) diet increases the abundances of *Clostridium* and *Ruminococcus* [55].

In the present study, the fecal microbiomes of FMT recipients fed raw food were enriched in *Bacteroides*, *Collinsella*, *Slackia*, and *Fusobacterium*. Similarly, dogs fed a Biologically Appropriate Raw Food (BARF) diet—a diet that consists of bones and raw meat with vegetables, fruits, and oil—harbor larger abundances of *Fusobacterium*, *Bacteroides*, and *Clostridium perfringens* compared to dogs fed a commercial diet [56]. Dogs switching from kibble to a diet of chicken meat and bone also exhibit increases in the relative abundances of *Collinsella*, *Enterococcus*, *Slackia*, *Faecalitalea*, and *Lactococcus* [57]. 

The impact of antibiotics on the fecal microbiomes of companion animals has been previously documented. Broad-spectrum antibiotics like tylosin, metronidazole, and amoxicillin can rapidly and significantly alter the microbiome composition and diversity of healthy and sick individuals [58,59,60]. These effects can be long-lasting and persist years after antibiotic administration. This study found that dogs that had taken antibiotics during the 12 months preceding sample collection had distinct fecal microbiomes compared to dogs that had not recently taken antibiotics. Specifically, they had an underrepresentation of *Allobaculum*, *Fusobacterium*, *Megamonas*, *Peptoclostridium*, and *Peptococcus*, and an overrepresentation of *Clostridiodes* and *Streptococcus* compared to dogs without an antibiotic exposure. This could be due to differences between bacterial species in their susceptibility (or resistance) to antibiotics and their ability to pump out, inactivate, or modify these bactericidal compounds [61,62,63]. Mechanisms of antibiotic resistance have been described for *Clostridioides* and *Streptococcus* species [64,65,66,67]. Genomic and proteomic analyses of metronidazole-resistant *C. difficile* isolates, for example, identified mutations in genes involved in electron transport (e.g., *glyC* and *nifJ*) [66,67] which altered redox potentials and the efficiency of antibiotic entry. Streptococci with reduced susceptibility to penicillin have mutations in genes coding for penicillin-binding proteins (PBPs) [64,65].

Lastly, we found that before FMT, dogs with higher body condition scores had more diverse fecal microbiomes than leaner dogs. Microbiome associations with body condition were also reported for a group of two-year-old Beagles [68], and several studies have demonstrated microbiome differences between lean and obese dogs [69,70]. However, the exact impacts of body condition on the microbiome are difficult to delineate given that that body condition is intertwined with a dog’s diet, breed, lifestyle, living environment, and health conditions. 

Interestingly, the fecal microbiomes of FMT recipients were not significantly predicted by clinical signs; that is, dogs with diarrhea did not have fundamentally different microbiomes from dogs with vomiting and/or constipation. This contrasts findings from a recently published study conducted in cats that took oral capsule FMTs for their chronic digestive issues [26]. That study reported that the fecal microbiomes of cats with diarrhea differed from the fecal microbiomes of cats with constipation and/or vomiting. These differences persisted two-weeks after FMT. This type of pattern was not observed in this dataset, potentially due to the stronger influences of other factors like diet and antibiotic use. 

### 4.3. Dynamics of Bacterial Engraftment in Oral FMTs

The present study reports that on average, 18% of the donor’s bacterial ASVs were transferred to the FMT recipient via the oral capsules. This was a slightly higher rate than was reported for cats that took these same FMT capsules, where only 13% of the donor’s ASVs transferred [26]. Similarly, 15% of Operational Taxonomic Units (OTUs) engrafted in the fecal microbiomes of humans with IBS [71]. A total of 15% of donor strains engrafted in humans undergoing FMT treatment for recurring *Clostridioides difficile* infection (rCDI) [27]. A meta-cohort study reported that only 10.8% of donor bacterial species were represented in FMT patients with rCDI, IBS, Crohn’s disease, and type 2 diabetes, among other diseases [72]. It is important to note that these engraftment efficiency rates are not directly comparable across studies, given that the methods for calculating the rates could differ.

We found that ASV engraftment rates were not associated with a donor’s identity or the diversity of the recipient’s starting microbiome, but were negatively associated with the diversity of the donor’s microbiome. This could simply be a product of how engraftment was calculated: richer donor microbiomes needed to engraft a larger number of ASVs than less diverse donors to achieve the same rates.

We did not find evidence that ASV engraftment rates were associated with any of the host factors examined (clinical signs, body condition score, prior antibiotic use, or diet). This contrasts a study conducted in humans with IBS, which found that an antibiotic pretreatment significantly reduced bacterial engraftment after FMT [71]. Podlesny et al. [72] found the opposite: antibiotic pretreatment increased engraftment in human patients with a range of diseases and treatment backgrounds.

Interestingly, the degree of similarity between the donor’s microbiome and the recipient’s starting microbiome influenced the abundance of donor ASVs in canine microbiomes after FMT. That is, for recipients that had microbiomes very similar to their donors, donor ASVs made up a smaller fraction of the microbiome. This could be due to donor bacteria competing with resident bacteria to occupy the same niches. In ecology, this is termed a “priority effect”, and describes when species that arrived earlier (or are already present) alter the resources or environmental conditions of species that arrive later (in this case donor bacteria) and affect their ability to establish in the community [73]. Less overlap between the donor’s microbiome and recipient’s microbiome might reduce the levels of bacterial competition or inhibition for engrafted strains. Nonetheless, a plethora of other factors such as the type of FMT [74], dosage/frequency of FMT, host genetics, and host health history are also thought to influence bacterial engraftment.

Engraftment rates were not uniform across bacterial species. Donors tended to share ASVs classified as uncultured *Lachnospiraceae*, *Lachnoclostridium*, *Ruminococcus*, *Fusobacterium*, and *Bacteroides*. Certain bacterial taxa like *Bacteroides* and *Fusobacterium* can be shared more easily because they are abundant in the microbiome of the donor. Yet other bacteria like *Alloprevotella*, *Prevotella*, and *Megamonas* were not as commonly shared despite being available for engraftment. Factors such as the metabolic flexibility and dietary specialization of the donor bacterium, their susceptibility to antibiotics and secondary metabolites, and their morphology (e.g., spore-forming, Gram-positive, or flagellated) could be at play. The interaction between the donor bacteria and the community of microbes already established in the FMT recipient could also be a strong determinant of which bacterial types will engraft.

Importantly, the engrafted ASVs were classified to genera and families of bacteria that may be playing important functions in gut health and homeostasis. *Lachnoclostridium* spp., uncultured *Lachnospiraceae* spp., and *Bacteroides* spp. for example, are key gut fermenters of dietary carbohydrates and protein [75,76,77], and in the process produce SCFAs that provide energy for colonic epithelial cells. Engrafted microbes such as *Bacteroides* spp. are also known to produce amines which are involved in the maintenance of mucosal homeostasis and the stability of DNA and proteins in host cells [78]. Some of these bacteria, like *Ruminococcus* spp. can degrade intestinal carbohydrates like mucin, which can support their growth and that of other gut bacteria [79,80]. Another important function these microbes may be providing is the hydrolysis of conjugated bile acids (BAs) into secondary BAs. Genomes belonging to *Lachnoclostridium,* for example, contain a gene cluster required for the multi-step dehydroxylation of BAs into SBAs [81]. This is significant given that the concentrations of secondary bile acids are significantly lower in dogs with CE compared to healthy dogs [8] and are hypothesized to inhibit the secretion of inflammatory cytokines. 

## 5. Limitations and Future Directions

This study has several limitations which impact how findings should be interpreted. Firstly, we did not include a placebo control group, and thus, we cannot describe with precision what effects could be attributed to the FMT capsules versus what effects are stochastic or due to other variables that the study did not measure. Secondly, the health conditions for some of the FMT participants were unknown or were not confirmed by veterinarians. This meant that we were not able to correlate microbiome responses with the dog’s actual health conditions. Related to this, this study had a highly heterogeneous pool of participants that came from different geographic areas and living situations, and had diverse diets, breeds, behaviors, health, and lifestyles. Patterns that were obscured in our dataset could emerge with a more defined experimental group. Lastly, the study employed 16S rRNA gene sequencing for profiling the microbiome, which gives limited species- and strain-level resolution and little insight into the functional capabilities of these microbes. Future studies should employ full-length 16S rRNA gene sequencing, shotgun metagenomics, or metabolomics for evaluating changes in the microbiome after fecal transplant. In particular, strain-level metagenomic analyses can identify with more precision and specificity the bacterial strains that are shared between donors and FMT recipients. 

## 6. Conclusions

Detailed investigations of microbiome responses to FMT in a sizable cohort of dogs are limited. Here, we report on the microbiome changes observed for 54 dogs that took oral capsule FMTs for their chronic digestive issues. We found that across participants, the relative abundances of five bacterial genera increased after FMT. Microbiome composition before and after FMT was modulated by host diet and prior antibiotic use. Lastly, we found that engraftment of donor bacteria is likely impacted by the similarity between the donor’s microbiome and the recipient’s microbiome before FMT. Our findings further our understanding of the factors potentially influencing microbiome responses and bacterial engraftment dynamics in dogs receiving oral capsule FMTs.

## Figures and Tables

**Figure 1 vetsci-11-00042-f001:**
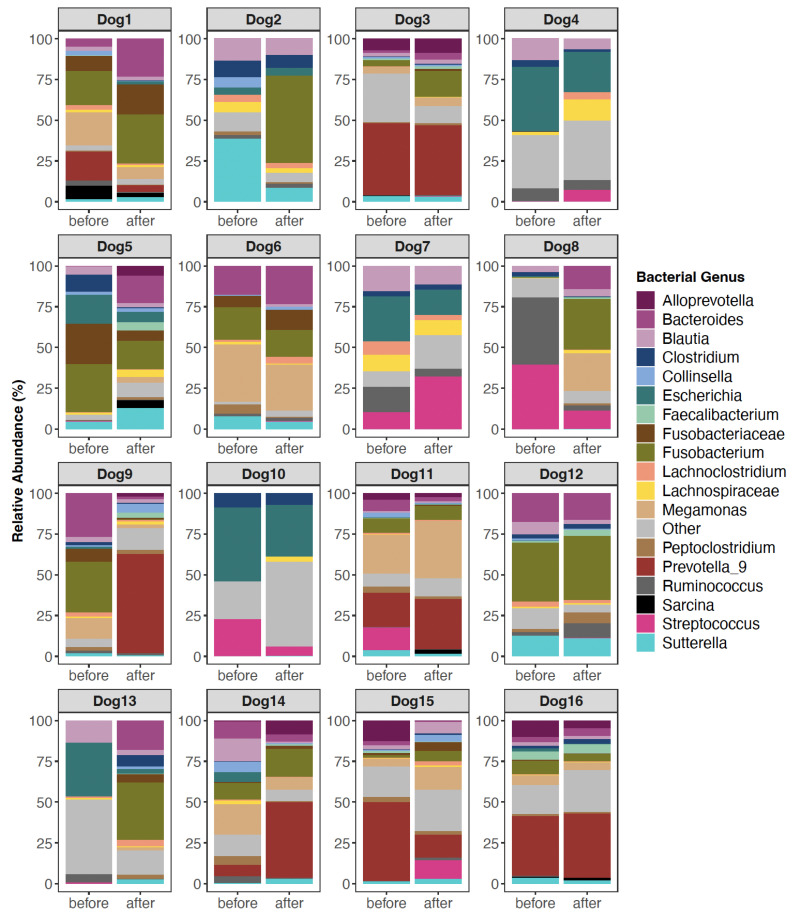
Fecal microbiome composition before and after FMT for a subset of canine participants. Only bacterial genera with mean relative abundances >1.3% are shown; the others are encapsulated by the “Other” category. Some sequences could not be classified to the Genus level, so their lowest known classification was used. For plots of all 54 dogs, see Appendix A.

**Figure 2 vetsci-11-00042-f002:**
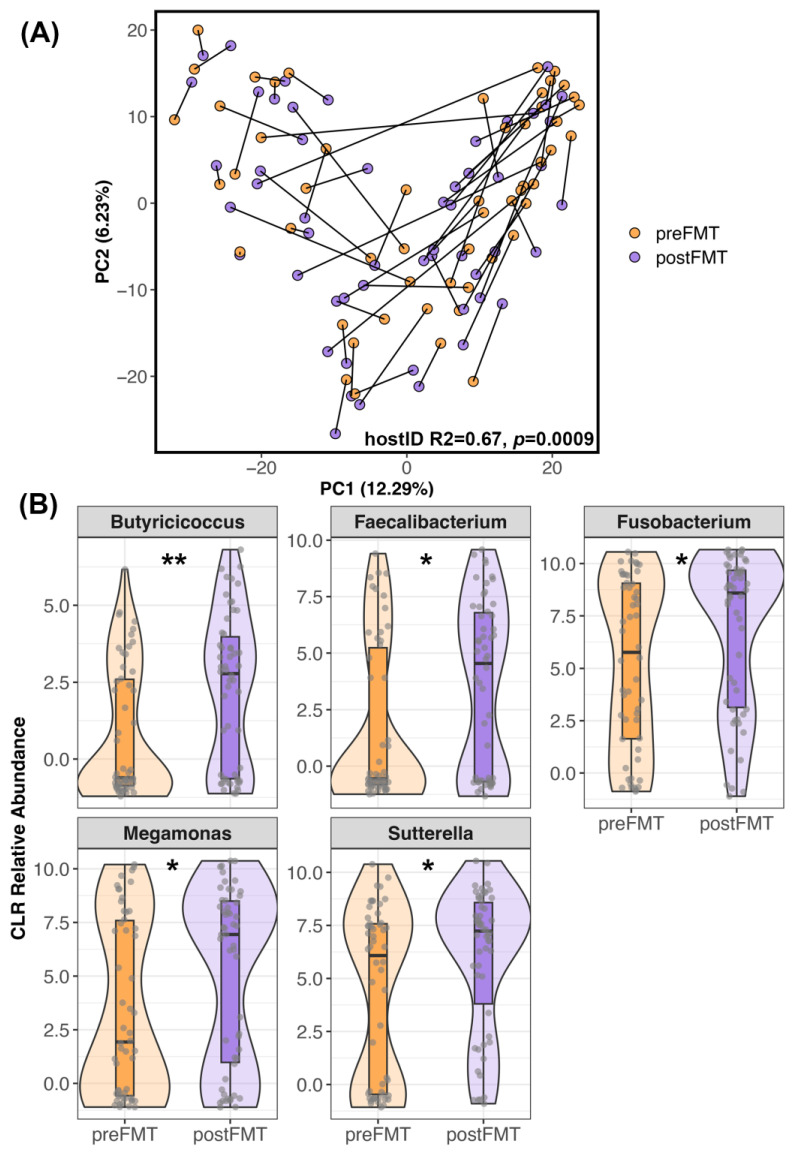
Differences in the microbiome after FMT. (**A**) PCoA ordination of FMT recipient microbiome samples before and after FMT. Samples from the same dog are connected. (**B**) Five bacterial genera were enriched in the fecal microbiomes of recipients after FMT compared to before FMT (*p* < 0.05, Appendix A). Center Log Ratio-transformed relative abundances are shown in the form of boxplots inside violin plots overlaid with a scatter of the individual data points. ** *p* < 0.01, * *p* < 0.05.

**Figure 3 vetsci-11-00042-f003:**
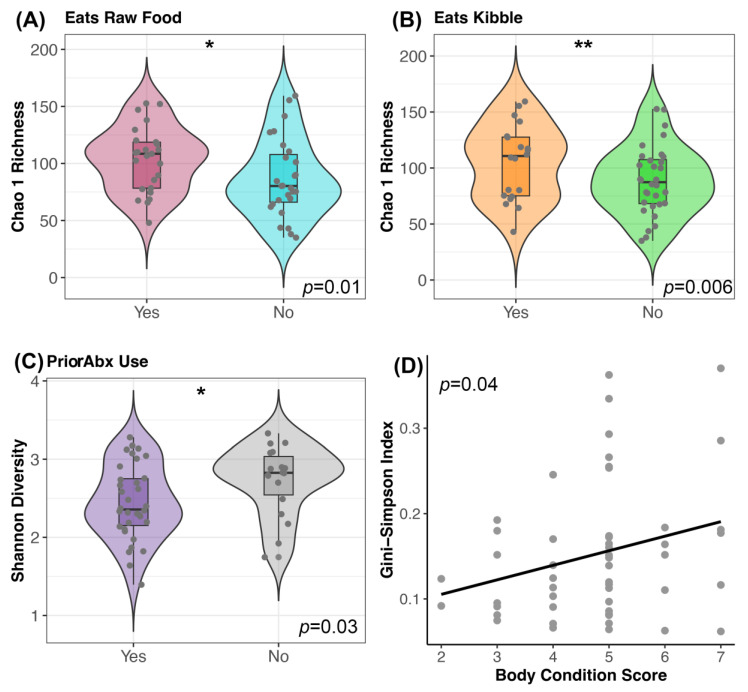
Host predictors of fecal microbiome alpha-diversity before FMT. Plots of microbiome diversity for FMT recipients before FMT, color coded by (**A**) raw food consumption (Yes vs. No), (**B**) kibble consumption (Yes vs. No) or (**C**) prior antibiotic use. For these plots, boxplots are inside violin plots which are overlaid with a scatter of the individual data points. ** *p* < 0.01, * *p* < 0.05 (**D**) Scatter of microbiome evenness by body condition score with a liner regression line. See Appendix A for associated statistics.

**Figure 4 vetsci-11-00042-f004:**
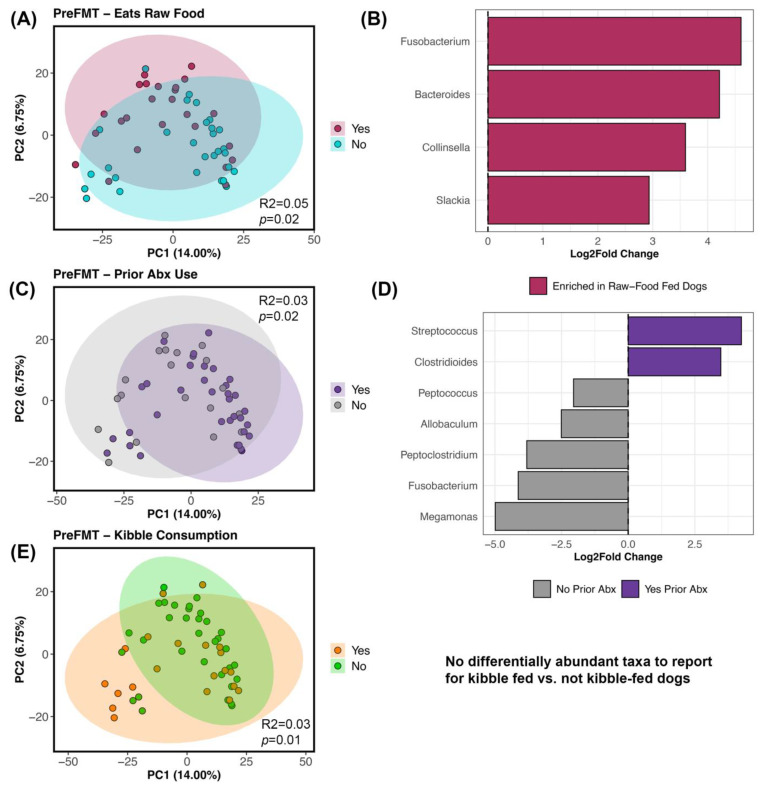
Host factors correlated with fecal microbiome beta-diversity before FMT. PCoA ordinations constructed from Bray–Curtis distances based on Genus-level bacterial abundances, color coded by (**A**) raw food consumption (Y/N), (**C**) prior antibiotic use (Y/N), and (**E**) kibble consumption (Y/N). (**B**,**D**) Differential abundance testing with R LinDA package to determine which bacterial genera differed in abundance between groups. See Appendix A for PERMANOVA statistics and Appendix A for LinDA statistics.

**Figure 5 vetsci-11-00042-f005:**
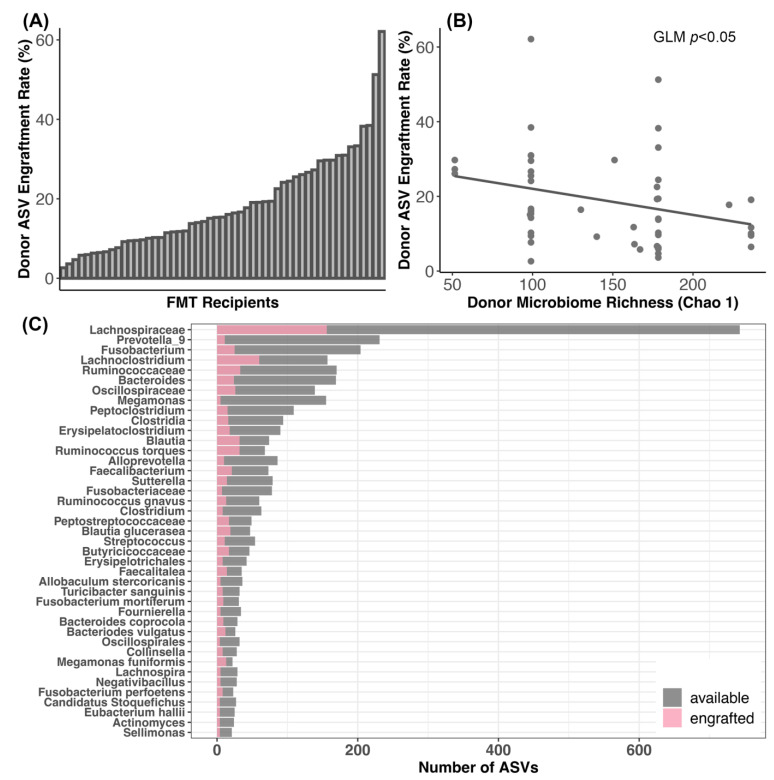
Engraftment of donor bacteria in FMT recipients. (**A**) Plots of donor bacterial amplicon sequence variant (ASV) engraftment rates across FMT recipients; 100% engraftment would indicate that all of the donor ASVs that could be shared were shared. (**B**) Relationship between ASV engraftment rates and donor microbiome alpha-diversity. (**C**) Taxonomic assignments of the donor ASVs most frequently shared with FMT recipients.

**Figure 6 vetsci-11-00042-f006:**
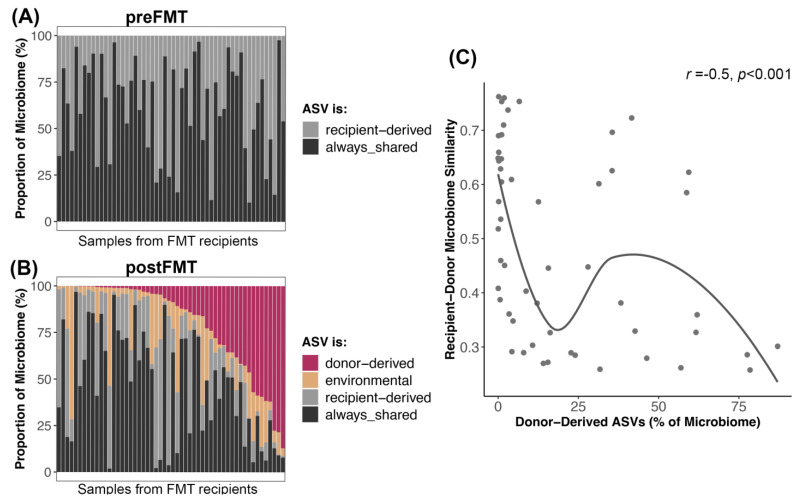
Microbiome reshaping after the introduction of donor-derived bacterial taxa. Bacterial ASVs present in FMT recipients post-FMT were categorized as donor-derived if they came from the donor, recipient-derived if they came from the recipient, always-shared if they were shared between donor and recipient before FMT, or as environmental/stochastic. (**A**,**B**) The proportion of the microbiome made up by these ASVs before and after FMT. (**C**) Abundance of donor-derived ASVs in postFMT microbiomes regressed against the microbiome similarity between donor and recipient preFMT. A smooth curve was overlaid.

**Table 1 vetsci-11-00042-t001:** Characteristics of the fifty-four dogs that took oral FMT capsules for their chronic digestive issues.

Characteristic	Specific Subcategory	N (%)
Age, in years	median & (range)	5.2 (1–15)
Body condition score *	median & (range)	5 (2–8)
Body weight category	<20 lbs	13 (24%)
	20–40 lbs	9 (17%)
	40–60 lbs	13 (24%)
	>60 lbs	19 (35%)
Sex	Female	25 (46%)
	Male	29 (54%)
Breed (broad)	Poodle	8 (15%)
	Golden Retriever	4 (7%)
	Terrier	6 (11%)
	German Shepherd	5 (9%)
	Other	31 (58%)
Diet * (not mutually exclusive)	Eat Kibble	21 (39%)
	Eat Raw Food	25 (46%)
	Eat Canned Food	27 (50%)
Spayed or Neutered	Yes	42 (78%)
	No	12 (22%)
Antibiotics *	Yes	35 (65%)
	No	19 (35%)
Initial clinical signs *	Diarrhea	26 (48%)
	Vomiting & Diarrhea	16 (30%)
	Vomiting	7 (13%)
	Constipation & Diarrhea	5 (9%)

Fifty-four dogs with chronic digestive issues (e.g., diarrhea, vomiting, and/or constipation episodes lasting >14 days) received oral capsule FMTs. Owners provided information on their health and lifestyle, and fecal samples were collected before and two-weeks after the end of a full course of capsules. Asterisks (*) distinguish the terms that were used in statistical models.

**Table 2 vetsci-11-00042-t002:** Summary of microbiome alpha-diversity and beta-diversity analyses for FMT recipients. An X indicates statistical significance (a = 0.05). See manuscript text or Appendix A for full statistical output.

	Alpha-Diversity		Beta-Diversity	
Predictor	preFMT	postFMT	preFMT	postFMT
Clinical signs				
Raw Food consumption	X		X	
Dry Food consumption	X		X	X
Prior antibiotics use	X		X	X
Body condition score	X			

## Data Availability

Please email the corresponding author for access to the Illumina 16S rRNA gene sequences for fecal samples included in this study. Tables containing ASV counts, ASV taxonomic labels, and sample metadata are accessible from this article (see Appendix A).

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
