# Peer review of "Microbiome Responses to Oral Fecal Microbiota Transplantation in a Cohort of Domestic Dogs"

_vetsci, 2024, doi:10.3390/vetsci11010042_

Round 1

Reviewer 1 Report

Comments and Suggestions for Authors

Well done. This is the kind of work I've been hoping to see from the AnimalBiome group - zero effort to "sell" their product, zero attempt to "wrap" a study around their product, but tremendous use of their capabilities to produce a really top notch study/publication that addresses a number of important and challenging issues in the FMT arena. There are some important limitations (eg. heterogeneity of cases), but the authors identify these, and there are a number of fascinating "contradictions" to some of the existing "dogma" (sorry for the pun), most of which have come from severely underpowered studies - perhaps the best part of this work is the number of important questions it raises - these are identified in a way that makes them amendable to scientific study, hence I hope to see more work from this group in the future.

Line 44 I would drop the "post-weaning" and just refer to it as parvovirus-associated diarrhea

Line 396 there's a major font change that I'm sure is unintentional

Line 405 Our study did not find evidence...

Author Response

Thank you for your thoughtful review. We have attached our responses in a Word document.

Reviewer 2 Report

Comments and Suggestions for Authors

The rationale for the research is appropriate as fecal microbiome transplants (FMT) in companion animals have not been extensively examined by many investigators. Although this is not a report of a tightly controlled study utilizing laboratory dogs from a captive population, it is a report of what could be considered a “real-world” situation. The experiments, statistics, and other analyses were performed to a high technical standard and are described in sufficient detail. The manuscript is well written and meets all applicable standards for the ethics of experimentation and research integrity.

This reviewer has only two minor comments. In line 109 it is reported fecal samples were stored at 48 degrees C. Is this the usual temperature for storage of fecal samples? Is there a reference for this procedure? One would think fecal samples should at least be stored refrigerated or in an ultracold freezer at -80C prior to extraction of DNA. The investigators assayed the FMT capsules for a variety of microbes (lines 118-120). Should the FMT material have also been assayed for the presence of Campylobacter spp.. Helicobacter spp., Clostridium piliforme, C. perfringens and possibly enteropathogenic Escherichia coli?

Author Response

(The authors gave the same response as above.)
